# Therapeutic Efficacy of Bromelain in Alveolar Ridge Preservation

**DOI:** 10.3390/antibiotics11111542

**Published:** 2022-11-03

**Authors:** Glauco Chisci, Luca Fredianelli

**Affiliations:** 1Department of Medical Biotechnologies, University of Siena, Policlinico “Le Scotte”, 53100 Siena, Italy; 2Institute of Chemical and Physical Processes of National Research Council, Via G. Moruzzi 1, 56124 Pisa, Italy

**Keywords:** extraction socket, alveolar bone remodeling, fresh socket, bromelain, pain, implant, bone resorption, bone remodeling, socket preservation, swelling

## Abstract

Most of research in regenerative oral surgery describes materials or techniques for increasing volumetric results for implant-supported prosthesis. The use of bio-materials in alveolar ridge preservation after tooth extraction commonly leads to a delayed recovery. Bromelain is an enzyme that belongs to a family of proteolytic enzymes derived from the stem of the pineapple plant (*Ananas comosus*) with effectiveness in decreasing the inflammation development and swelling. The present paper reports a prospective comparative study performed in order to test the possible use of oral bromelain 40 mg in alveolar ridge preservation. Evaluations were performed at three time points after the surgery: after 2 days (t1), after 7 days (t2) and after 14 days (t3). A statistically significant difference among patients that used bromelain and patients that used placebo resulted among the use of bromelain and lower Visual Analogue Scale (VAS) at t1 (r = −0.75, *p* = 0.0067), t2 (r = −0.90, *p* = 0.0001) and t3 (r = −0.8566, *p* = 0.0008). Bromelain therapy reported a statistically significant difference among patients that used bromelain and patients that used placebo even with regards to the use of bromelain and postoperative swelling at t1 (r = −0.79, *p* = 0.0034), t2 (r = −0.81, *p* = 0.0020) but not at t3 (r = −0.34, *p* = 0.2967). With the result of the present paper, and the poorness of contraindication of the investigated drug, bromelain may be suggested to be used for patients that undergo to alveolar ridge preservation after tooth extraction.

## 1. Introduction

Particular bone volume represents a great matter of interest for implant dentistry.

An appropriate bone volume in an edentulous patient lead to a correct implant placement and correct implant-supported prosthodontics.

Rapidly after tooth extraction, in fact, the alveolar bone around the removed tooth undergoes a change, a modification of its volume leading to a three-dimensional resizing that arises few days after the extraction and goes ahead for many months. This would lead to shrinkness, where half of the bone volume is reported to be resorbed during the initial period of repair after the extraction [1].

Research by Tan et al. [2] in 2012, and Van der Weijden et al. [3] in 2009, reported that the alveolar socket undergoes a mean horizontal resorption between 0.9 and 3.8 mm and an average vertical shrinkness of 1.24 mm, within 3 to 7 months after tooth extraction: this alveolar boner reduction leads to a modification of level of soft tissue too.

Evidence of this bone resorption is more evident with an horizontal reduction of the buccal bone wall mostly in its coronal aspect: the presence of bone atrophy in an edentulous site is characterized by a difficulty in a correct implantology or by the need for guided bone regeneration (GBR) before or contextual to the implant surgery, in addition to the probable need to increase the soft tissues [4]. The placement of a correct implant is the first step for correct implant prosthodontics. For this reason, while immediate implant insertion after tooth extraction seems to jeopardize the vertical bone remodeling of the socket, the technique of alveolar ridge preservation (ARP) after tooth extraction represents a challenge for the surgeon in order to preserve a correct bone volume for implant surgery [5,6]: the possibility to influence the modulation of a correct bone volume after tooth extraction represents the basis for correct implant dentistry. For many years, the main treatment for alveolar bone atrophy has been the GBR or incrementation with bone onlay [7,8]: the ratio of the ARP is the prevention of bone shrinkness and following bone atrophy after tooth extraction: in this way, a correct implantology could be possible without further interventions. However, the use and the placement of bio-materials inside the alveolar socket for the ARP after tooth extraction commonly leads to a delayed recovery, as this material interferes in part with biology [9]. Delayed recovery in oral surgery is commonly reported by patients with worsted levels of health-related quality of life [10]. For this reason, in order to reduce infectious complications and indirectly the swelling, antibiotics prescription is commonly used with local or systemic administration [11]: for this purpose, the use of anti-inflammatory drugs is often replaced by antibiotics [12,13].

Bromelain is an enzyme that belongs to a family of proteolytic enzymes derived from the stem of the pineapple plant (*Ananas comosus*). The chemical structure of bromelain is reported in Figure 1.

The most remarkable characteristic is its effectiveness in decreasing the inflammation development and swelling. The mechanisms of action of bromelain appears to be clear, as the compound is capable of inhibiting the formation of bradykinin thanks to the depletion of the plasma Kallikrein system Bromelain also seems to reduce leukocyte migration into inflamed areas by removing the CD128 chemokine receptors, and creating a barrier for adhesion of leukocytes to blood vessels at the site of inflammation [14].

The present paper investigates a possible role of oral administration of bromelain 40 mg in ARP. The possible use of this drug in ridge preservation is presented and discussed with the aim of reducing pain and swelling in patients scheduled for ARP.

## 2. Materials and Methods

The present work reports a prospective study performed to compare possible benefits of bromelain (Bromelina ANANASE ANGELINI 40 mg anhydrous dibasic calcium phosphate, monohydrate lactose, dihydrate dibasic calcium phosphate, Macrogol 4000, cornstarch, Colloidal silica, Talc, Magnesium stearate, Stearic acid, Eudragit L30 D-55, Triethyl citrate, Simethicone, Gelatin, Opalux AS-23014 (Sucrose, Quinoline Yellow, Titanium Dioxide, Orange Yellow S, Povidone, Sodium Benzoate) Wax powder, Sucrose) in alveolar ridge preservation. A total of 22 patients (18 males; 4 females) were enrolled in the present study. Age ranged from 28 to 67 years old, with mean 46.8 +/− 3.2 years standard deviation.

Eligible patients (inclusion criteria) were selected among those older than 18 years old, systemically healthy, with a diagnosis of decayed lower first molar and indication to tooth extraction and ARP and delayed implant surgery, with an example reported in Figure 2. Patients were excluded if they: (i) requiring anticoagulation therapy (ii) had systemic diseases that could interfere with oral tissue healing process/bleeding (iii) were using bisphosfonates (iv) were pregnant (v) had mental/physical disabilities (vi) underwent radiation treatment to the head or neck region (vii) infection of the interested tooth (viii) periodontitis (ix) received antibiotic therapy in the last month.

Patients who were willing to participate were asked to sign a written informed consent, in which treatment planning was discussed and benefit/risk ratio was explicated, with agree for participation in the present study, for processing of personal data and images, and for publishing purposes, approved by the Istitutional Review Board (scientific ethical committee of centro dentistico Chisci, Grosseto, Italy, 17 September 2019). All study procedures complied with the principles stated in the Declaration of Helsinki “Ethical Principles for Medical Research Involving ‘Human Subjects’”, adopted by the 18th World Medical Assembly, Helsinki, Finland, June 1964, and as amended most recently by the 64th World Medical Assembly, Fortaleza, Brazil, October 2013.

None of the patients referred the habit of smoke, and no pathological health conditions were present. All the interventions were performed by the same surgeon. Under local anesthesia with 1:100,000 articaine without the use of a flap the tooth was gently extracted with luxation and pliers; the alveolar socket was smoothened and cleansed with irrigation of 0.9 NaCl for 30 s. All patients after received a sponge of deproteinized bovine bone + collagen 100 mg covered with resorbable collagen membrane 16 mm × 22 mm stabilized with 6 6/0 resorbable sutures around the alveolar socket. An example is reported in Figure 3.

Sutures were removed at 21 days after surgery.

A randomization protocol was produced for the distribution of patients in the two treatment groups. Treatment assignments were stored in sealed envelopes and opened at the time of surgery. Clinical staff who were not involved in the surgery procedure performed the allocation sequence: half of the patients were selected as test group and received 2 capsules of bromelain 40 mg once per day 2 days before and 7 days after surgery. The control group instead received paracetamol 1000 g administration once per day for 7 days after surgery. Postoperative pain with Visual analogue scale (VAS) and postoperative mandibular swelling were measured in all patients. VAS was measured with a visual scale (1–10) where 1 was the lower value and 10 the higher value: patients were asked to judge their pain with a single number from 1 to 10. Postoperative swelling was measured with the increase of a mean value of two lines, the first one was traced from mandibular angle-subnasion, the second one was traced from mandibular angle-lip as sketched in Figure 4.

Measurements were performed before surgery (t0) and at three time points after the surgery: 2 days after surgery (t1); 7 days after surgery (t2); 14 days after surgery (t3).

Association between variables was evaluated with univariate regression analysis; *p*-values < 0.05 were considered statistically significant.

## 3. Results

All extraction sites recovered uneventfully with no complications; no additional surgical intervention or administration of additional medications resulted needed. Data regarding VAS score in test group and control group are reported in Table 1, while data regarding swelling score in test group and control group are in Table 2.

Patient that used bromelain reported lower VAS values at all the three times: a statistically significant correlation resulted among the use of bromelain and lower VAS values at t1 (r = −0.75, *p* = 0.0067), t2 (r = −0.90, *p* = 0.0001), t3 (r = −0.8566, *p* = 0.0008), as shown in Figure 5. As the same for VAS, patients with bromelain therapy reported a reduced swelling after the intervention at the first and second time: the group of bromelain reported a statistically significant correlation at t1 (r = −0.79, *p* = 0.0034), t2 (r = −0.81, *p* = 0.0020) but not at t3 (r = −0.34, *p* = 0.2967), as shown in Figure 6.

The group of patients that used bromelain reported anyway better results during the recovery: this aspect was confirmed both on the basis of the evaluations of the VAS in the following days and on the basis of the evaluation of the swelling (Figure 5 and Figure 6).

## 4. Discussion

The use of bio-materials in oral surgery in commonly related to delayed recovery, with worsened health related quality of life [10]. ARP is a technique that has been studied extensively on the basis of biomaterials and on the basis of surgical approach, with or without a flap, resulting in a better postoperative course in the case of a flapless surgery [15]: the elevation of the flap for tooth extraction leads commonly to a worsened recovery with compromised blood supply for the clot and probably an increased bone resorption [15]. The use of drugs for postoperative management of pain and swelling after ARP little has been studied in the literature, giving priority to evaluate biomaterials in ARP: for this reason, the novelty of our study is to examine a matter of interest for many patients and researchers.

The properties of bromelain have been studied over the years to understand if it may have a contribution in the prevention and treatment of negative symptoms after the extraction of mandibular third molars. de la Barrera-Núñez et al. in their paper reported no statistically significant difference among the use of bromelain and placebo, reporting anyway lower symptoms in bromelain group [16]; Bormann et al. in their article reported good results for bromelain although with no statistically significant difference [17]; Ghensi et al. in their randomized controlled trial reported a moderate anti-inflammatory efficacy of bromelain, reducing postoperative swelling, albeit not to any significant extent compared with no drug administration: further they reported best results with the combined use of bromelain and dexamethasone sodium phosphate in terms of control of postoperative discomfort [18].

At the time of writing the present paper, in the authors’ knowledge no articles have studied the efficacy of bromelain in ARP: the novelty of present study is the intention to reduce this discomfort for the patients that undergo to ridge preservation.

Commonly in oral and maxillofacial surgery submucosal cortisone injections were considered useful to reduce postoperative pain and swelling [19]: however, the authors’ opinion is that cortisone efficacy in reducing pain and swelling would be offset by the effect of reduction of tissue mineralization proper of this drug, leading to a marked bone resorption and jeopardizing the possibility of further implantology.

The present study was then conceived and based on experience in third molar surgery and scientific support of the use of bromelain in this field [20]: for this reason, in the present work the authors theorized that bromelain could have a role in the delayed recovery after ARP. The results have shown a better recovery for the patients that used bromelain in alveolar ridge preservation. The reduction of pain and swelling in this procedure is a favorable outcome, as many patients complain about this delayed recovery (compared to simple tooth extractions) and most or the research in this field is mainly interested in volume preservation than patients health related quality of life. Patients that used bromelain reported significantly fewer swelling results and reduced VAS pain scores.

The limitations of the present study were probably the methods of measurement of facial swelling: our method was defined on the basis of the research of Osunde et al. [21], but this method led to some possible bias. Amin and Laskin in 1983 defined probably a better measurement method based on two distances, the first from corner of mouth to attachment of ear lobe, while the second from outer canthus of eye to angle of mandible [22]: further research in the field of swelling measurements in oral and maxillofacial surgery could benefit from the use of facial scanners, as reported by Bormann et al. [17].

Bromelain is a light drug with poor contraindications: Zehra and Tashfeen in their review reported that bromelain equals non-steroidal anti-inflammatory drugs as an anti-inflammatory agent: in Europe it is approved for oral and topical use, mainly for surgical wounds, inflammation due to trauma and surgery, and debridement of deep burns [23]. Further bromelain was commonly suited for patients due to lack of compromise in its peptidase efficacy and the absence of undesired side effects [24]: it has been reported as having positive effects on the respiratory, digestive, and circulatory systems, and potentially on the immune system [25]. In fact, recent research underlined a potential preventive value of the synergistic effects of bromelain with curcumin against severe COVID-19 [26].

Reduced discomfort after alveolar ridge preservation leads to a better quality of life. While many papers in literature report benefit for third molar surgery, this work reported a statistically significant role of bromelain in reduce the postoperative pain and swelling.

On the basis of our small sample of results, this paper suggests to consider the use of bromelain for patients that undergo to ridge preservation after tooth extraction: further study should investigate possible benefits of bromelain in others regenerative interventions, as alveolar bone augmentation and sinus lift.

## 5. Conclusions

Alveolar ridge preservation is an intervention that aims to preserve a correct bone volume for implant surgery. In the present article, pain and swelling in 22 patients were evaluated after tooth extraction and ARP. Patients that used bromelain reported better results after ridge preservation in terms of swelling and pain, compared to control patients at 2 (t1) and 7 days (t2) after surgery, and lesser pain at 14 days (t3) too.

## Figures and Tables

**Figure 1 antibiotics-11-01542-f001:**
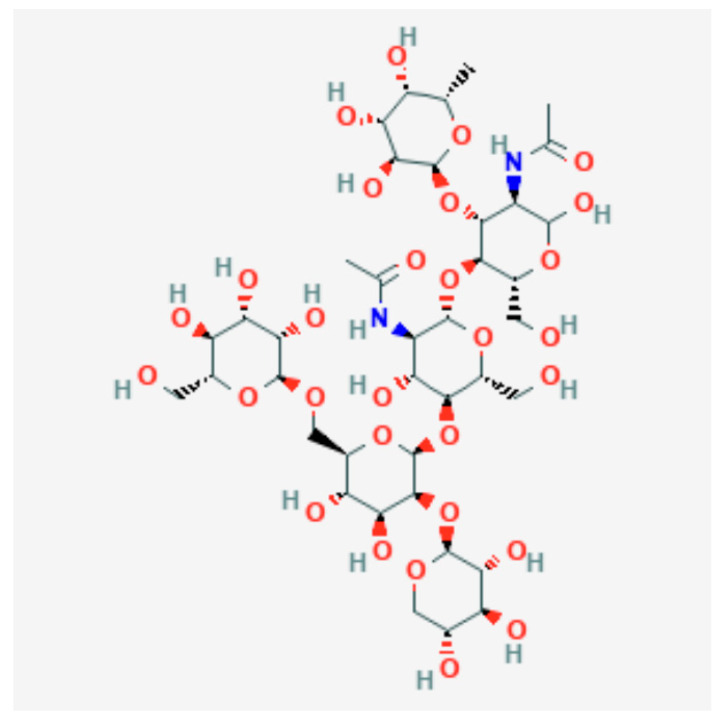
Chemical structure of bromelain. National Center for Biotechnology Information (2022). PubChem Compound Summary for CID 74981710, Bromelain. Retrieved 30 September 2022 from https://pubchem.ncbi.nlm.nih.gov/compound/Bromelain (accessed on 27 October 2022).

**Figure 2 antibiotics-11-01542-f002:**
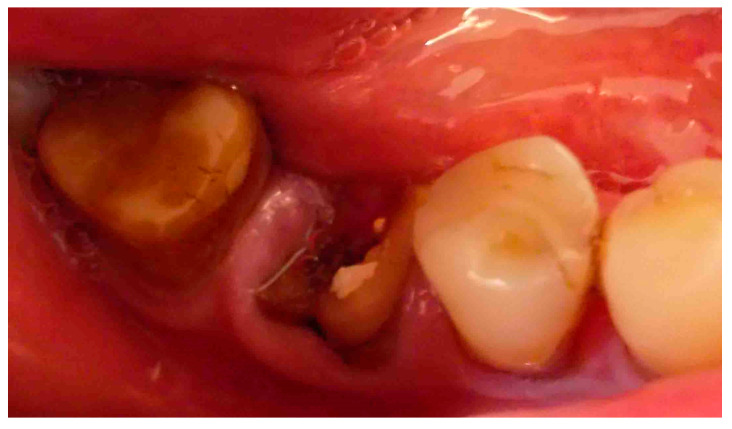
Roots of a decayed mandibular first molar (46) that received indication of extraction and alveolar ridge preservation.

**Figure 3 antibiotics-11-01542-f003:**
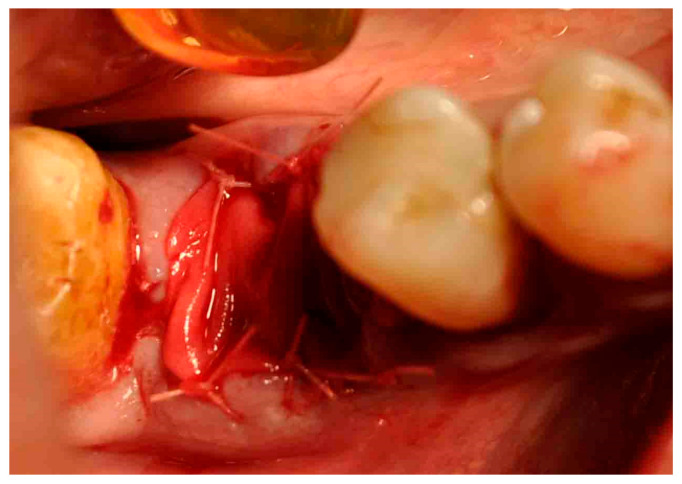
Postoperative socket with sponge of deproteinized bovine bone + collagen 100 mg covered with resorbable collagen membrane 16 mm × 22 mm stabilized with 6 6/0 resorbable sutures.

**Figure 4 antibiotics-11-01542-f004:**
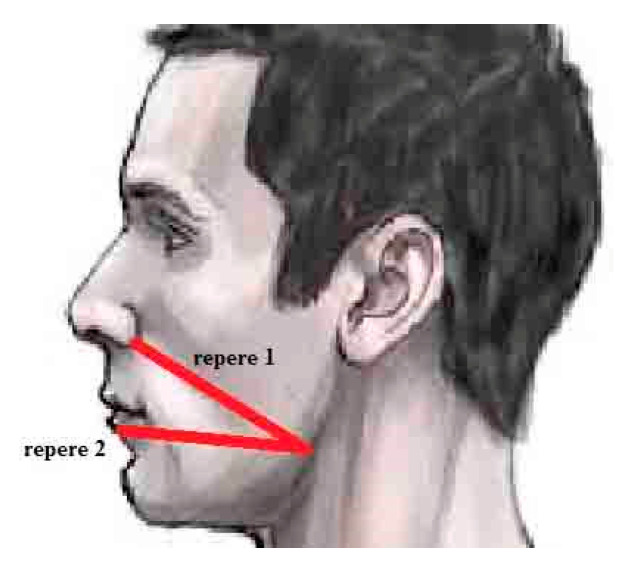
Referrals on the face for the measurement of the swelling.

**Figure 5 antibiotics-11-01542-f005:**
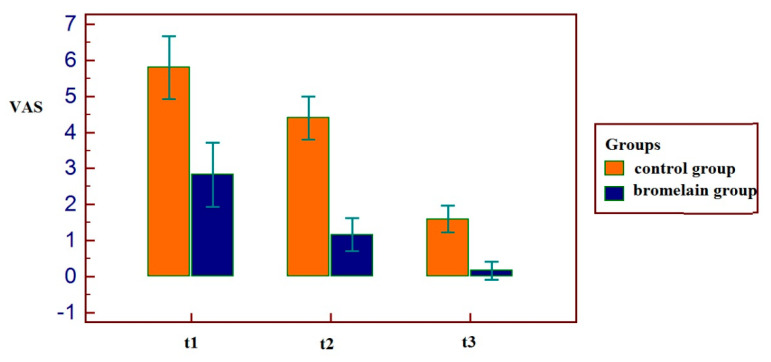
Decrease of pain measured with VAS in all patients. Lower results are observable in patients with bromelain prescription.

**Figure 6 antibiotics-11-01542-f006:**
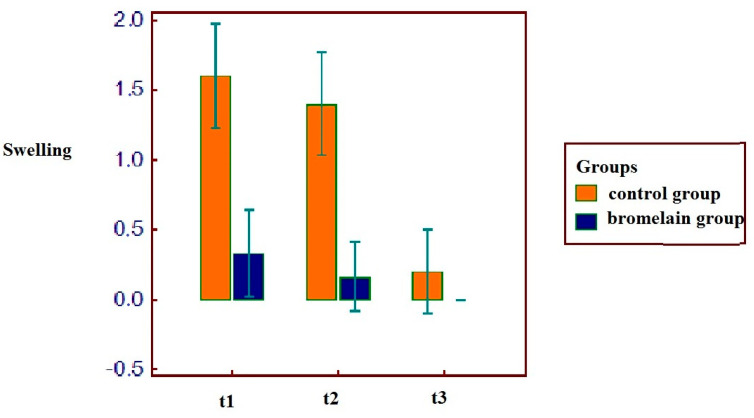
Decrease of swelling in all patients. Lower results are observable in patients with bromelain prescription.

**Table 1 antibiotics-11-01542-t001:** VAS score in test group and control group.

	Swelling t1 (Mean Value +/− Standard Deviation)	Swelling t2 (Mean Value +/− Standard Deviation)	Swelling t3 (Mean Value +/− Standard Deviation)
Bromelain group	0.3 +/− 0.5	0.2 +/− 0.4	0.0 +/− 0.0
Control group	1.6 +/− 0.5	1.4 +/− 0.5	0.2 +/− 0.4

**Table 2 antibiotics-11-01542-t002:** Swelling score in test group and control group.

	VAS t1 (Mean +/− Standard Deviation)	VAS t2 (Mean +/− Standard Deviation)	VAS t3 (Mean +/− Standard Deviation)
Bromelain group	2.8 +/− 1.4	1.1 +/− 0.7	0.2 +/− 0.4
Control group	5.8 +/− 1.2	4.4 +/− 0.8	1.6 +/− 0.5

## Data Availability

Not applicable.

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
