# Peer review of "Therapeutic Efficacy of Bromelain in Alveolar Ridge Preservation"

_antibiotics, 2022, doi:10.3390/antibiotics11111542_

Round 1

Reviewer 1 Report

The authors present an article that is easy to understand, however they do not add much to the existing scientific literature about the substance in question.

 The article has a series of problems in the writing, scientific and methodology used in the study. The whole article needs a serious correction of the English.

Other points to consider:

Abstract

Ø  Modify the Abstract to make it more precise, ex: “The present paper  reports a prospective comparative study performed in order to test the possible use of oral bromelain 40 mg in alveolar ridge preservation.” Is this the aim of the work?

Ø  It isn`t “ three different times - it is “time points”

Ø  A statistically significant correlation resulted among the use of bromelain and lower Visual Analogue Scale (VAS) at t1 (r=-0.75, 16 p=0.0067), t2 (r=-0.90, p=0.0001) and t3 (r=-0.8566, p=0.0008)” this is not scientifically correct, should be explained

Ø  With the result of the present paper, and the poorness of contraindication of the investigated drug, bromelain may be suggested in patients with reported pain and swelling after tooth extraction.this is not the tittle of the paper” Pain and swelling was never mentione before. And the alveolar ridge preservation?

Introduction

Ø  International literature [2, 3] has reported …….”the authores have reported

Ø  “Most of research in ARP aimed to describe materials or techniques to obtain volumetric increased results ……” .It is not perceptible

Ø  Recente research…….[10-12]”. Is it recent? 2014?2016?

Ø  nanas comosus or Ananas comusus

Ø  The chemical structure of bromelain is reported in Figure 1.  Is this a figure from the authors? Or taken from a site?

o   National Center for Biotechnology Information (2022). PubChem Compound Summary for CID 74981710, Bromelain. Retrieved September 30, 2022 from https://pubchem.ncbi.nlm.nih.gov/compound/Bromelain.

Ø  The reference 13 dates to 1967. Isn`t there any study more recente?

Ø  Bromelain also seems to reduce leukocyte migration into inflamed areas by removing the CD128 chemokine receptors, and creating a barrier for adhesion of leukocytes to blood vessels at the site of inflammation.” Where is the reference?

Ø  ARP or ARD??????

Materials and methods ( needs a serious refining)

Ø  Why it was used Bromelain 40 mg? Why this dose? Is it pure?

Ø  Where is the control group? Did they take nothing? Paracetamol?

Ø  Are the inclusion and exclusion criteria enough? I think not

Results

-How did they do the VAS? Reported by the patient? Can they use medium in the VAS score?

Discussion

Ø  It`s very “messy” and confusing. The authors talk about implants, scaffolds and platelet rich fibrin all at once trying a  comparison to a natural substance- bromelain. They should talk about the chemical, immunological and other propreties of the Bromelain, that reinforce its use.

Ø  HRQL ??????

Conclusions

Ø  ARB?

Ø  Should be more simple

In the

Institutional Review Board Statement: “IRB approval was not required as bromelain is commonly used in dentistry and is not an experimental drug” If this is true, should be mentioned in the introduction. Do dentists use it frequently everywhere? Is this natural supplement not used only in CVD and burns debridment?

Author Response

The authors wish to thank the reviewers for their time and efforts spent on our paper in order to improve its quality. All the suggestions have been considered and applied when possible, together with the other reviewer ones and the editor’s too.

  1. The authors present an article that is easy to understand, however they do not add much to the existing scientific literature about the substance in question. The article has a series of problems in the writing, scientific and methodology used in the study. The whole article needs a serious correction of the English.

Many thanks to the reviewer: the entire manuscript received a full correction of the English.

  1. Other points to consider: abstract

Modify the Abstract to make it more precise, ex: “The present paper  reports a prospective comparative study performed in order to test the possible use of oral bromelain 40 mg in alveolar ridge preservation.” Is this the aim of the work?

The purpose of this study in the abstract was correctly reported

  1. Ø It isn`t “ three different times”  - it is “time points”

The authors thank the reviewer for having pointed out this typo, which has now been corrected.

  1. “A statistically significant correlation resulted among the use of bromelain and lower Visual Analogue Scale (VAS) at t1 (r=-0.75, 16 p=0.0067), t2 (r=-0.90, p=0.0001) and t3 (r=-0.8566, p=0.0008)” this is not scientifically correct, should be explained

Many thanks to the reviewer: the concept was rephrased and correctly explained

  1. “With the result of the present paper, and the poorness of contraindication of the investigated drug, bromelain may be suggested in patients with reported pain and swelling after tooth extraction.this is not the tittle of the paper” Pain and swelling was never mentione before. And the alveolar ridge preservation?

Many thanks to the reviewer: the concept was rehprased and correctly explained

  1. Introduction “International literature [2, 3] has reported …….”the authores have reported

Thanks to the reviewer, the correct term was used as suggested.

  1. “Most of research in ARP aimed to describe materials or techniques to obtain volumetric increased results ……” . It is not perceptible

Thanks for pointing out the issue. The authors fully rephrased this concept in order to explain better the purpose of the ARP.

  1. “Recent research…….[10-12]”. Is it recent? 2014?2016?

We agree with reviewer’s opinion and this paragraph was fully rephrased.

  1. nanas comosus or Ananas comusus

The authors thank the reviewer for having pointed out this typo, which has now been corrected.

  1. The chemical structure of bromelain is reported in Figure 1. Is this a figure from the authors? Or taken from a site?

The figure was taken from pubchem and reported in the caption.

National Center for Biotechnology Information (2022). PubChem Compound Summary for CID 74981710, Bromelain. Retrieved September 30, 2022 from https://pubchem.ncbi.nlm.nih.gov/compound/Bromelain.

  1. The reference 13 dates to 1967. Isn`t there any study more recent?

Thanks to the reviewer’s suggestion, a new and more recent references has been added.

  1. “Bromelain also seems to reduce leukocyte migration into inflamed areas by removing the CD128 chemokine receptors, and creating a barrier for adhesion of leukocytes to blood vessels at the site of inflammation.” Where is the reference?

The authors thank the reviewer for having pointed out this typo, which has now been corrected by adding the missing ref.

  1. ARP or ARD??????

The authors thank the reviewer for having pointed out this typo, which has now been corrected.

  1. Materials and methods ( needs a serious refining) Why it was used Bromelain 40 mg? Why this dose? Is it pure?

The Bromelain 40mg was chosen as in our country it is a over the counter medication ((Bromelina ANANASE  40 mg) this product contains Calcio fosfato bibasico anidro, Lattosio monoidrato, Calcio fosfato bibasico biidrato, Macrogol 4000, Amido di mais, Silice colloidale, Talco, Magnesio stearato, Acido stearico, Eudragit L30 D-55, Trietilcitrato, Simeticone, Gelatina, Opalux AS-23014 (Saccarosio, Giallo di chinolina, Diossido di titanio, Giallo arancio S, Povidone , Benzoato di sodio) Cere in polvere, Saccarosio. The name of the drug was reported in the paper.

  1. Where is the control group? Did they take nothing? Paracetamol?

Many thanks to the reviewer for the indication to underline the evidence of the control group, this information was reported.

  1. Are the inclusion and exclusion criteria enough? I think not

The inclusion criteria and exclusion criteria paragraph were fully rephrased.

  1. Results

-How did they do the VAS? Reported by the patient? Can they use medium in the VAS score?

Patients were asked to judge their pain with a single number from 1 to 10, this information was reported in the paper.

  1. Discussion

It`s very “messy” and confusing. The authors talk about implants, scaffolds and platelet rich fibrin all at once trying a  comparison to a natural substance- bromelain. They should talk about the chemical, immunological and other propreties of the Bromelain, that reinforce its use.

Many thanks to the reviewer for the suggestion. The discussion was fully rephrased with the removal of the references and argument pointed out by the reviewer and reported concept about chemical, immunological and other properties of Bromelain.

  1. HRQL ??????

The authors thank the reviewer for having pointed out this typo, which has now been corrected.

  1. Conclusions

ARB?

The authors thank the reviewer for having pointed out this typo, which has now been corrected.

  1. Should be more simple

We agree with the reviewer suggestion, the conclusions were modified as requested.

In the Institutional Review Board Statement: “IRB approval was not required as bromelain is commonly used in dentistry and is not an experimental drug” If this is true, should be mentioned in the introduction. Do dentists use it frequently everywhere? Is this natural supplement not used only in CVD and burns debridment?

Many thanks to the reviewer for this precision: IRB approval was not required for experimental test as  this drug is commonly used in Italy: this natural supplement is often used in medicine and dentistry. This concept was reported in the introduction as requested. Approval for the patient consent, processing of personal data and images and for publishing purposes was reported in the methods.

Reviewer 2 Report

Dear Authors,

your topic is of grat interest in oral surgery. However, some deficiencies must be improved.

- Your in vivo study must report the ethical commitee acceptance code, and you have to clarify the Helsinki declaration. 

- The introduction must be improved. The use of antinflammatory drugs is sometime replaced by antibiotics in order to avoid infectious complications and indirectly the edema. Since you are describing an Italian study, you may add some citation about it such as: 

D'Agostino S, Dolci M. Antibiotic therapy in oral surgery: a cross sectional survey among Italian dentists. J Biol Regul Homeost Agents. 2020 Jul-Aug,;34(4):1549-1552. doi: 10.23812/20-183-L. PMID: 32867465. 

Rodríguez Sánchez F, Arteagoitia I, Rodríguez Andrés C, Caiazzo A. Antibiotic prophylaxis habits in oral implant surgery among dentists in Italy: a cross-sectional survey. BMC Oral Health. 2019 Dec 2;19(1):265. doi: 10.1186/s12903-019-0943-x. PMID: 31791306; PMCID: PMC6889412.

- You have to add the standard deviation for each mean value you show. 

- Your method to perform the sweelling measurements has some bias. I wonder why you did not use the Amin MM. et al. (1983) method. You may read about it with this cite: Amin MM, Laskin DM. Prophylactic use of indomethacin for prevention of postsurgical complications after removal of impacted third molars. Oral Surg Oral Med Oral Pathol. 1983 May;55(5):448-51. doi: 10.1016/0030-4220(83)90227-x. PMID: 6575332.

- You have to clarify the randomization method used. 

- You have to clarify the absence of antibiotic prophilaxys/therapy among inclusion criteria.

- You have to clarify why only 5/22 patients receved the test therapy.

- You have to clarify the specific characteristic for Bromelain (brand, excipients).

Author Response

The authors wish to thank the reviewers for their time and efforts spent on our paper in order to improve its quality. All the suggestions have been considered and applied when possible, together with the other reviewer ones and the editor’s too.

  1. Dear Authors,

your topic is of great interest in oral surgery. However, some deficiencies must be improved.

- Your in vivo study must report the ethical commitee acceptance code, and you have to clarify the Helsinki declaration.

Many thanks to the reviewer for the suggestion, the Helsinki declaration and the approval for the patient consent, processing of personal data and images and for publishing purposes with date of the scientific commitee was reported in the methods; IRB approval for bromelain test was not required as in our country bromelain is an over the counter drug often used in medicine and dentistry.

  1. The introduction must be improved. The use of antinflammatory drugs is sometime replaced by antibiotics in order to avoid infectious complications and indirectly the edema. Since you are describing an Italian study, you may add some citation about it such as:

D'Agostino S, Dolci M. Antibiotic therapy in oral surgery: a cross sectional survey among Italian dentists. J Biol Regul Homeost Agents. 2020 Jul-Aug,;34(4):1549-1552. doi: 10.23812/20-183-L. PMID: 32867465.

Rodríguez Sánchez F, Arteagoitia I, Rodríguez Andrés C, Caiazzo A. Antibiotic prophylaxis habits in oral implant surgery among dentists in Italy: a cross-sectional survey. BMC Oral Health. 2019 Dec 2;19(1):265. doi: 10.1186/s12903-019-0943-x. PMID: 31791306; PMCID: PMC6889412.

Many thanks to the reviewer, this concept was reported in the paper and the references were cited.

  1. You have to add the standard deviation for each mean value you show.

The standard deviation was clarified for each mean value.

  1. Your method to perform the sweelling measurements has some bias. I wonder why you did not use the Amin MM. et al. (1983) method. You may read about it with this cite: Amin MM, Laskin DM. Prophylactic use of indomethacin for prevention of postsurgical complications after removal of impacted third molars. Oral Surg Oral Med Oral Pathol. 1983 May;55(5):448-51. doi: 10.1016/0030-4220(83)90227-x. PMID: 6575332.

Many thanks to the reviewer suggestion: this aspect was reported in the discussion with our referral and the citation war reported in the reference.

  1. You have to clarify the randomization method used.

The authors wish to thank the reviewer for having pointed out such an omission. The randomization method was reported.

  1. You have to clarify the absence of antibiotic prophilaxys/therapy among inclusion criteria.

The inclusion criteria and exclusion criteria paragraph were fully rephrased and this concept was reported.

  1. You have to clarify why only 5/22 patients received the test therapy.

Many thanks to the reviewer, the number reported was wrong and the correct definition with the randomization method used was reported.

  1. You have to clarify the specific characteristic for Bromelain (brand, excipients).

Many thanks to the reviewer for the suggestion, this information was reported in the paper. (Bromelina ANANASE  40 mg) this product contains Calcio fosfato bibasico anidro, Lattosio monoidrato, Calcio fosfato bibasico biidrato, Macrogol 4000, Amido di mais, Silice colloidale, Talco, Magnesio stearato, Acido stearico, Eudragit L30 D-55, Trietilcitrato, Simeticone, Gelatina, Opalux AS-23014 (Saccarosio, Giallo di chinolina, Diossido di titanio, Giallo arancio S, Povidone , Benzoato di sodio) Cere in polvere, Saccarosio.

Round 2

Reviewer 1 Report

In this second round the authors now provide a much cleaner and solid information about the work they done ( both in the Introduction and Discussion). The aim is more clear as the methods used to reach it. The references were updated, the convoluted texto disappeared and ethical concerns were also cleared also.The english editing is much better.

Minor mistakes:

When referring to “64th World Medical 114 Assembly, Fontaleza, Brazil, October 2013.” Is Fortaleza not Fontaleza ( Materials and Methods and Institutional Review Board Statement)

In discussion 231-24- format the size of the letter

Author Response

Point by point manuscript ID: antibiotics-1962616

Reviewer 1

  1. When referring to “64th World Medical 114 Assembly, Fontaleza, Brazil, October 2013.” Is Fortaleza not Fontaleza ( Materials and Methods and Institutional Review Board Statement)

Many thanks for your kind suggestion, the term was corrected.

  1. In discussion 231-24- format the size of the letter

Apologize for the inconvenience, which has been now fixed.

Reviewer 2 Report

Dear Authors,

please find the following corrections.

- The brand used for Bromelain must be followed by the registred symbol, the specific characteristics must be translated in English.

- the standard deviation has to be written right after the average value, without words in between.

- Fig 4. Has a wrong English word for tooth.

- Line 98 has to be re-written for the age.

- Line 101: Requiring? or required?

- Line 107: You reported the Scientific Committee, but since your study ia a clinical trial, you MUST have the approval of a ufficial Ethical Committee.

Author Response

Point by point manuscript ID: antibiotics-1962616

Reviewer 2

  1. The brand used for Bromelain must be followed by the registred symbol, the specific characteristics must be translated in English.

The brand and registered symbol of the bromelain was inserted with the translated characteristics.

  1. the standard deviation has to be written right after the average value, without words in between.

Many thanks for the suggestion, the standard deviation was reported as requested.

  1. Fig 4. Has a wrong English word for tooth.

Many thanks to the reviewer, the term was corrected.

  1. Line 98 has to be re-written for the age.

The text was rephrased for a better comprehension.

  1. Line 101: Requiring? or required?

The text was rephrased.

  1. Line 107: You reported the Scientific Committee, but since your study ia a clinical trial, you MUST have the approval of a ufficial Ethical Committee.

Many thanks for your kind suggestion, the term was corrected and IRB approval reported.